# The Moderating Effect of Self-Construal on the Relationship Between Mindfulness and Forgiveness

**DOI:** 10.3390/bs15020195

**Published:** 2025-02-12

**Authors:** Zhiruo Guo, Ting Xu, Haijiang Li

**Affiliations:** 1School of Psychology, Shanghai Normal University, Shanghai 200234, China; 0364473@sd.taylors.edu.my (Z.G.); tingx_k@163.com (T.X.); 2Lab for Educational Big Data and Policymaking (Ministry of Education), Shanghai Normal University, Shanghai 200234, China

**Keywords:** forgiveness, mindfulness, interdependent self-construal, independent self-construal

## Abstract

The extant literature has established an association between mindfulness, forgiveness, and self-construal; however, the mechanisms underlying these relationships are not yet clear. The present study aims to explore the potential moderating role of self-construal between mindfulness and forgiveness. Four hundred and nineteen participants were recruited and asked to complete measures of mindfulness, self-construal, state, and trait forgiveness. The results showed that mindfulness was positively correlated with state and trait forgiveness. The present study investigated the association between mindfulness and state forgiveness, as well as trait forgiveness, and how this was moderated by self-construal. The findings indicated that for participants who identified with interdependent self-construal, both associations increased, while for those who identified with independent self-construal, the relationship decreased or became non-significant. These results align with previous research and suggest that the mechanisms identified may be applicable to psychological education. The results of this study suggest the presence of a potential underlying mechanism between mindfulness and forgiveness through self-construal. The findings of the research have guiding significance for the design and implementation of psychological interventions, providing evidence that mindfulness training may promote forgiveness by affecting self-construal. These findings have the potential to enhance interpersonal relationships, promote cross-cultural communication and international cooperation, and cultivate students’ mindfulness, empathy, and forgiveness, thereby supporting the creation of a more harmonious and supportive learning environment.

## 1. Introduction

Conflict and offense are inevitable components of interpersonal relationships ([72]). Beyond the natural human inclination to respond to offenses with anger or a desire for retribution, individuals can also exercise their will to forgive offenders ([35]). Forgiveness is a complex psychological construct influenced by various factors, with prosocial tendencies as its primary characteristic ([68]). Interpersonal forgiveness has been shown to effectively heal wounds, restore trust, and mend damaged relationships. Forgiveness can be categorized into two distinct types: state forgiveness, which is situation-specific, and trait forgiveness, which reflects an individual’s general propensity to forgive across different contexts ([45]).

The repercussions of forgiveness have been demonstrated to extend to mental health, physical health, and social relationships. In terms of mental health, forgiveness has been linked to a decrease in symptoms of depression and anxiety, as well as an increase in life satisfaction ([59]). On a physiological level, forgiveness has been shown to reduce the risk of cardiovascular diseases, alleviate chronic pain, and enhance overall health ([40]). Finally, in terms of social relationships, forgiveness is crucial for conflict resolution and the maintenance of healthy interpersonal connections ([52]).

Promoting individual forgiveness is crucial. Numerous studies have identified various factors that influence the forgiveness process, including psychological traits, sociocultural backgrounds, and personal experiences. Psychological factors such as emotional regulation, self-construal, empathy, and mindfulness are significant predictors of forgiveness ([15]). Individuals exhibiting higher levels of mindfulness are more likely to demonstrate forgiving behaviors ([72]). Cultural differences also play a role in the propensity to forgive, with individuals from collectivist cultures more inclined to forgive for the sake of social harmony ([60]). Therefore, we will explore the relationship between two variables, mindfulness and self-construal, and forgiveness in depth. Meanwhile, trying to find effective ways to promote forgiveness.

### 1.1. Relationship Between Mindfulness and Forgiveness

A substantial body of research has explored the relationship between mindfulness and forgiveness. Several studies have identified a positive correlation between mindfulness and the granting of forgiveness ([37]). Furthermore, [1] ([1]) demonstrated that mindfulness can mediate the association between forgiveness and health. In another study, after eight weeks of mindfulness practice, participants reported decreased levels of stress and rumination, along with increased levels of forgiveness ([1]). Additionally, [34] ([34]) observed improved mood and attitudes toward forgiveness following mindfulness training compared to a matched control group. However, other studies have produced contradictory findings, indicating that mindfulness training may not significantly affect forgiveness behaviors ([61]). The impact of mindfulness on forgiveness appears to be influenced by other factors, such as psychoeducation related to forgiveness ([5]). In our study, we try to investigate whether self-construal could also influence the relationship between mindfulness and forgiveness.

### 1.2. Relationship Between Self-Construal and Forgiveness

A considerable body of research has demonstrated a correlation between self-construal and forgiveness. The interdependent self, characterized by a relational existence, is hypothesized to exhibit a greater propensity for forgiveness compared to the independent self. The theoretical foundations of forgiveness, as elucidated by the concept of self-construal, facilitate a comprehensive understanding of this phenomenon. However, real-life scenarios reveal controversies surrounding the relationship between self-construal and forgiveness. Prior studies have indicated a positive correlation between interdependent self-construal and specific forms of forgiveness ([43]; [32]). Forgiveness contributes to the harmony and stability of interpersonal relationships ([33]). In contrast, [70] ([70]) found that individuals with low interdependent self-construal exhibited a greater propensity for retaliation against transgressors. Furthermore, researchers have posited that independent self-construal, through self-face concern, is negatively associated with forgiveness, while interdependent self-construal, through other-face concern, is positively associated with forgiveness in both the United States and China ([71]). [43] ([43]) demonstrated that trait forgiveness serves as a mediator in the relationship between self-construal and well-being. Interdependent self-construal encourages individuals to engage in forgiveness behaviors to maintain relationship harmony, although these behaviors may not necessarily extend to emotional forgiveness ([32]).

### 1.3. Relationship Between Self-Construal and Mindfulness

The relationship between self-construal and mindfulness is an area of research that has been explored in the academic literature. Self-construal is one of the most robust and reliable predictors of social goals and behaviors ([55]). The central tenet of mindfulness is the cultivation of self-awareness ([55]). [26] ([26]) found that mindfulness exerts a significant indirect influence on happiness through the clarity of self-concept. The capacity for mindfulness to enhance self-awareness is attributed to its ability to encourage engagement with the present moment, thereby facilitating a more accurate self-concept by reducing biased perceptions ([65]). Specifically, mindfulness has been shown to be positively associated with self-referential processing ([6]), self-enhancement, and self-concept ([55]; [26]). [55] ([55]) suggested that brief mindfulness induction led to a decrease in prosocial behavior among individuals with relatively independent self-construal and an increase among those with interdependent self-construal.

### 1.4. Current Study

Previous studies have found that mindfulness can promote individual health and well-being and has a significant positive correlation with forgiveness. However, self-construction is a cognitive tendency of individual self-knowledge, and no research has investigated the relationship between mindfulness and forgiveness from this perspective. Therefore, this study aims to explore the interrelationships among mindfulness, self-construal, and forgiveness, given their close connections.

Based on previous research, we found the practice of mindfulness has been shown to effectively reduce the impact of negative emotions by reducing rumination, enhancing emotional awareness, and promoting cognitive reappraisal ([17]; [37]). Consequently, this can result in the promotion of prosocial behaviors such as forgiveness. A substantial corpus of empirical studies lends support to this proposition, demonstrating that individuals who possess higher levels of mindfulness are more inclined to exhibit forgiving behavior ([37]). Furthermore, mindfulness training has been demonstrated to enhance an individual’s capacity for forgiveness by reducing stress, enhancing self-awareness, and promoting positive emotions ([14]). This study delineates two distinct forms of forgiveness: state forgiveness and trait forgiveness. Based on these findings, H1 is proposed.

**Hypothesis (H1).** 
*Mindfulness has a positive relationship with trait and state forgiveness.*


Dependent self-construal is predicated on the premise that there exists a relationship and interdependence between individuals and others. Individuals place greater emphasis on social harmony and interpersonal relationships. In the event of offense, these individuals demonstrate a propensity to seek redress and sustain relationships through forgiveness, a tendency that has been positively correlated with trait forgiveness ([19]). Conversely, individuals with independent self-construal are more likely to engage in retaliatory or avoidant behaviors ([32]). The practice of mindfulness has been shown to facilitate more effective management of emotional responses and promote prosocial behaviors, such as forgiveness, by enhancing awareness and acceptance of current emotions ([23]; [17]). Individuals with dependent self-construal may benefit more from mindfulness practice because they pay more attention to social harmony and interpersonal relationships, and mindfulness practice can further enhance their forgiveness behavior ([55]). Conversely, individuals with independent self-construal may not experience equivalent benefits from mindfulness practice as they prioritize personal goals and self-realization. Consequently, the effect of mindfulness practice on their forgiveness behavior may be less pronounced ([55]). Hence, H2 is proposed.

**Hypothesis (H2).** 
*Self-construal moderates the relationship between mindfulness and trait (state) forgiveness.*


## 2. Literature Review

### 2.1. Mindfulness

Mindfulness is defined as an individual’s habitual level of awareness and attention to the present moment ([12]; [17]; [36]) or, alternatively, as a state of heightened engagement and awakening to the present ([58]). The psychological dimensions of mindfulness, including compassion, forgiveness, loving-kindness, and empathy, are significant components of mindfulness training. Compassion in mindfulness meditation is characterized by the deliberate cultivation of a heartfelt response to the suffering of others, with the intention of alleviating that suffering ([16]). It is a central aspect of many contemplative practices and is associated with enhanced well-being and prosocial behavior ([16]).

Two key elements can be extracted from the definition of mindfulness ([10]; [37]): (a) the monitoring of individual internal experiences, including thoughts, emotions, and bodily sensations occurring in the present moment, and (b) the conscious attention paid to these experiences with an open, objective, and non-judgmental attitude. Individuals who have committed crimes often revel in their traumatic experiences ([37]), leading to rumination and the cultivation of negative emotions and thoughts, such as anger and revenge ([37]; [47]; [57]), which hinder forgiveness ([37]; [7]). Consequently, incorporating mindfulness practices has the potential to enhance well-being by reducing stress and negative emotions, thereby promoting prosocial behaviors ([17]; [24]; [14]; [34]; [66]; [8]). Furthermore, mindfulness has been shown to curtail rumination (e.g., anger and hostility), enabling individuals to disengage from negative emotions and re-evaluate events ([37]; [34]; [28]). This may, in turn, facilitate forgiveness.

### 2.2. Forgiveness

For many years, people have held divergent opinions on the definition of forgiveness. However, with the advancement of research in this field, these disagreements have gradually diminished, leading to a broad consensus among scholars ([69]). [18] ([18]) proposed a three-aspect model of forgiveness, encompassing cognition, emotion, and behavior. This positive transformation, according to [18] ([18]), occurs in the three aspects of cognition, emotion, and behavior following an offense. [45] ([45]) elucidated forgiveness from the perspective of motivation transformation, positing that forgiveness is a process of change in prosocial motivation. This process entails a shift in an individual’s motivations, with negative sentiments such as desire for revenge, avoidance, and alienation toward the offender being gradually replaced by positive motivations toward the offender. It is important to note that [46] ([46]) emphasized that forgiveness is not a specific motivation but rather a process of motivation transformation. This transformation is reflected in the victim replacing negative behavior with positive behavior, thereby forming prosocial motivation.

Forgiveness is a positive process and an active choice to deal with harm. It has been argued that forgiveness is not indicative of weakness or an act of submission ([20]). In order to comprehend the concept of forgiveness, it is imperative to distinguish it from related concepts, such as forgetting, tolerance, and reconciliation, in order to avoid any potential confusion. Firstly, it is important to note that forgiveness and forgetting are not synonymous. Although the details of the harmful incident may gradually become blurred over time, the painful emotions caused by the intense harm may still exist. Secondly, forgiveness and forgetting are not compatible. The process of forgiveness can only occur when the victim is clearly aware of the existence of the harm ([20]). Secondly, forgiveness is also different from tolerance. When individuals choose to tolerate harmful behavior, they tend to downplay the harm they have suffered or even think that these harms are insignificant. This approach may appear more straightforward than confronting and processing their distressing emotions. Nevertheless, it is only when an individual possesses sufficient strength of character that they are capable of honestly acknowledging their suffering and processing these emotions in a rational manner, thereby attaining authentic forgiveness. Consequently, forgiveness should not be regarded as a process that suppresses personal development; rather, it should be recognized as a significant catalyst for personal growth ([49]). Finally, it is important to note that forgiveness and reconciliation are two distinct processes. Forgiveness does not necessarily require reconciliation to occur concurrently, and the two processes can be carried out independently ([69]).

### 2.3. Self-Construal

Self-construal is defined as an individual’s self-perception of generalized self-image traits. This concept can be categorized into two distinct types: independent and interdependent self-construal. These categories are associated with individualistic and collectivist cultures, respectively ([71]). The independent self-construal conceptualizes the self as a distinct and autonomous entity whose behavior is less susceptible to the influence of others. In contrast, interdependent self-construal places significant emphasis on interpersonal dynamics ([19]). Individuals exhibiting a stronger tendency toward interdependent self-construal often describe themselves in relation to the influence of others, such as family or community.

[44] ([44]) advanced a two-dimensional conceptualization of self-construal, distinguishing between independent and dependent self-construal. This seminal study delved into the multifaceted nature of self-construal, exploring its dimensions across a range of domains, including definition, structure, salient characteristics, tasks, sources of self-concept, and the basis of self-esteem. Independent self-construal emphasizes the separation of individuals from the social environment, and the structure of the self is clearly defined, single, and stable; the main task of dependent individuals is to gain a sense of belonging, find a position that suits them, and take appropriate actions to support others in achieving their goals. Self-construal is widely regarded as one of the most robust and reliable indicators for predicting social goals and behaviors ([44]). Individuals who exhibit independent self-construal demonstrate a heightened concern for the realization of personal interests and objectives. In contrast, individuals who manifest dependent self-construal exhibit a heightened focus on collective interests and a stronger emphasis on the maintenance of harmonious interpersonal relationships ([31]). However, research findings indicate that individuals possess both independent and dependent self-components. Self-construal can be conceptualized as both a stable trait and a dynamic state that can be activated by situations or induced experimentally ([21]).

### 2.4. Emotion Regulation Theory

Emotion regulation theory is a seminal framework within the domain of psychology, which is utilized to elucidate the mechanisms by which individuals modulate and regulate their own emotional states ([11]). The theoretical foundation of this framework is predicated on the pursuit of emotional balance, the mitigation of the impact of negative emotions, and the enhancement of positive emotions through the implementation of cognitive and behavioral strategies ([50]). The broad spectrum of emotion regulation can be categorized into two distinct types: cognitive emotion regulation and behavioral emotion regulation. Cognitive emotion regulation primarily entails cognitive processing of emotional events, such as reinterpreting events through cognitive reappraisal or regulating emotional expression through expression inhibition ([63]). Conversely, the second type of emotion regulation, i.e., behavioral emotion regulation, involves the regulation of emotions through practical actions, such as seeking social support or relaxation training. The emotion regulation process model proposed by [23] ([23]) divides emotion regulation into two categories: regulation before emotion generation and regulation after emotion generation. This model emphasizes the dynamics and timing of the process. The dual process model of emotion regulation further differentiates between automatic and controlled regulation processes, thereby unveiling the intricacy and multi-level character of emotion regulation ([25]).

Emotion regulation theory has been extensively applied across numerous disciplines ([53]). In the domain of mental health, the ability to regulate emotions effectively has been demonstrated to play a pivotal role in averting the onset of psychological maladies such as anxiety and depression. In the domain of education, emotion regulation training has been shown to enhance students’ academic performance and mental well-being ([56]). In the context of interpersonal relationships, effective emotion regulation facilitates the cultivation of positive and fulfilling relationships. Finally, in the field of organizational behavior, the ability of employees to regulate their emotions is a significant factor in their job satisfaction and commitment to the organization ([41]).

## 3. Materials and Methods

### 3.1. Participants

The present study’s sample primarily comprises college students from a university in eastern China, with a total of 419 participants initially recruited for the study. Following the exclusion of 11 participants who provided atypical answers or outlier responses (exceeding the mean by three standard deviations), the final valid sample size was 408. The sample comprised 303 women (74.3%) and 105 men (25.7%), with an average age of 20.30 years (SD = 2.53 years). Participants were selected based on two criteria: full-time undergraduate status and voluntary participation following comprehension of the study design and content. The sample was randomly selected. The recruitment of participants was achieved through a multifaceted approach. Primarily, class leaders were tasked with disseminating information regarding the study amongst their respective classes, ensuring the reach of all students and facilitating voluntary participation. Additionally, the school’s student affairs management system and social media platforms were utilized as conduits for recruitment, thereby expanding the scope of this study. The screening criteria excluded students who did not meet the age and educational background requirements, as well as participants who filled in incomplete or untrue information. The questionnaires were distributed from 1 April to 30 April 2024 and collected from 1 May to 15 May 2024. The measurement items used for data collection are shown in Appendix A.

This study ensured the representativeness of the sample and the reliability of the data through a variety of recruitment methods and strict screening criteria. The samples exhibited relatively consistent educational and cultural backgrounds, thereby reducing potential interference factors caused by background differences. However, this also limits the general applicability of the research conclusions. It is recommended that future studies consider expanding the sample range to include individuals of different ages, educational backgrounds, and cultural backgrounds in order to enhance the external validity of the study.

### 3.2. Measures

#### 3.2.1. Mindfulness Scale

The trait mindfulness level of the participant was measured using a Chinese version of the Mindful Attention Awareness Scale (MAAS) complied by [12] ([12]). The MAAS has 15 items, using a six-point scale ranging from “almost always” to “rarely”. The scale had high internal consistency reliability (α = 0.890), retest reliability (r = 0.870), and validity ([42]). In this study, the α coefficient of the scale was 0.86.

#### 3.2.2. Trait Forgiveness Scale

The trait forgiveness level of the participant was measured using a Chinese version of the Trait Forgiveness Scale (TFS) compiled by [9] ([9]). TFS has 10 items, using a four-point scale ranging from “never” to “always”, where questions 1, 3, 6, 7, and 8 are reverse-scored. The higher the overall score on the scale, the higher the level of trait forgiveness of the individual. The scale has a high degree of internal consistency confidence (α = 0.75) and validity ([4]). In this study, the scale had an α coefficient of 0.65.

#### 3.2.3. State Forgiveness Scale

The motivation for interpersonal aggression of the participant was measured using a Chinese version of the Transgression-Related Interpersonal Motivation Scale (TRIM-12) compiled by [48] ([48]). TRIM-12 has 12 items, including 5 questions for the retaliation dimension and 7 questions for the avoidance dimension, using a five-point scale ranging from “completely disagree” to “completely agree”. Higher scores obtained for the retaliation and avoidance dimensions indicate lower levels of state forgiveness. For the explanatory purpose, the reverse scoring of the state forgiveness score was adapted; that is, the higher the total score of the questionnaire, the higher the level of state forgiveness in an individual. The scale has high internal consistency reliability (α = 0.87), retest reliability (r = 0.79), and validity ([39]). In this study, the α coefficient for this scale was 0.87, and the α coefficients for the dimensions of retaliation and avoidance were 0.83 and 0.89, respectively.

#### 3.2.4. Self-Construal Scale

The trait of self-construal of the participant was measured using a Chinese version of the Self-construal Scale (SCS) compiled by [64] ([64]), and translated by [51] ([51]). SCS has 16 items, including 10 questions in the interdependent self-dimension and 6 questions in the independent self-dimension, using a seven-point scale ranging from “completely disagreed” to “completely agreed”. Referring to the processing method of [31] ([31]), the average score of the independent self-dimension and interdependent self-dimension were calculated and subtracted to obtain the individual self-construal index. The higher the score is, the more inclined the individual is to independent self-construal; otherwise, the more inclined the individual is to interdependent self-construal. The scale has a high degree of internal consistency confidence (α = 0.88), retest reliability (r = 0.87), and validity ([51]). In this study, the scale had an α coefficient of 0.74 and an α coefficient of 0.73 and 0.58 for the two dimensions, respectively.

In this study, we utilized several validated scales to assess mindfulness, trait forgiveness, state forgiveness, and self-construal. Mindfulness was gauged by the Chinese version of the Mindfulness Attention Awareness Scale (MAAS), which exhibited an internal consistency reliability (Cronbach’s α) of 0.86, indicative of adequate construct validity and criterion validity. Trait forgiveness was measured by the Chinese version of the Trait Forgiveness Scale (TFS), with an internal consistency reliability of 0.65, which, although slightly lower than the ideal value, remains sufficiently practical for educational applications. State forgiveness was measured by the Chinese version of the Deviance-Related Interpersonal Motivation Scale (TRIM-12), with a total α coefficient of 0.87, demonstrating high internal consistency. Self-construal was measured by the Chinese version of the Self-Construct Scale (SCS), with a total α coefficient of 0.74, an α coefficient of 0.73 for the independent self-construal dimension, and an α coefficient of 0.58 for the dependent self-construal dimension, showing good construct validity and criterion validity. Additionally, the single-factor variance explanation rate was 12.44%, which was lower than 40%, indicating that there was no serious common method bias. The reliability and validity of these scales ensured the scientificity and credibility of the research results and provided a solid foundation for evaluating the relationship between mindfulness, trait forgiveness, state forgiveness, and self-construal. The measurement items used for data collection are shown in Appendix A.

Even though individual scales have been shown to have low reliability in research contexts, in educational practice, the philosophy of pragmatism emphasizes the practical application and functionality of tools and methods. As [29] ([29]) noted, practicality is of crucial importance in educational practice. Consequently, the value of tools and scales in educational practice is determined not by their theoretical perfection but by their practical efficacy in achieving the established educational objectives and needs.

### 3.3. Statistical Analysis

SPSS23.0 statistical analysis software was used to analyze the questionnaire data on mindfulness, trait forgiveness, state forgiveness, and self-construal. Furthermore, the moderating effects were examined using the PROCESS macro vs. 4.1 in SPSS26.0. We used 5000 bootstrapped iterations. Standardized estimation was used to conduct moderation analyses, taking into account variations in scoring methods across different scales ([27]).

## 4. Results

### 4.1. Descriptive Statistics and Bivariate Associations

The means, standard deviations, range, and correlations for all variables are presented in Table 1. The results of correlations show that trait mindfulness, trait forgiveness, and state forgiveness exhibited significantly positive correlations with one another (*p*s < 0.01). Self-construal showed a significantly positive correlation with trait mindfulness (*p* < 0.01). Trait forgiveness had a positive correlation with independent self-construal, and interdependent self-construal had a positive correlation but did not have significant correlations with state forgiveness or trait forgiveness (*p*s > 0.01).

### 4.2. Common Method Bias Analysis

To rule out the effects of using the same kind of instrument on the results, a common method of bias analysis was conducted. As the previous studies suggested, the total variance for a single factor is less than 40%, and there is no serious common method bias ([54]). The results revealed that the variance explanation rate was 12.44%, so there is no common method bias in the present study.

### 4.3. Moderation Analyses

#### 4.3.1. The Moderating Role of Self-Construction Between Trait Mindfulness and Trait Forgiveness

The relationships between trait mindfulness, self-construal, and their interaction with trait forgiveness were explored using Model 1 in PROCESS macro ([27]). The results of the analysis predict the trait forgiveness with mindfulness level (β = 0.20, t = 3.86, *p* = 0.001), self-construal index (β = −0.11, t = −2.18, *p* = 0.030), and their interaction (β = −0.100, t = −1.98, *p* = 0.048).

The interaction was further investigated using the procedures of [2] ([2]) by analyzing simple slopes for the two respects self-construal and plotting them. At low levels of the self-construal index, specifically interdependent self-construal, mindfulness was significantly positively related to trait forgiveness (β = 0.30, t = 4.16, *p* < 0.001). While this association was not significant at high levels of the self-construal index, specifically independent self-construal (β = 0.09, t = 1.23, *p* > 0.05) (Figure 1).

#### 4.3.2. The Moderating Role of Self-Construction Between State Mindfulness and Trait Forgiveness

The relationships between mindfulness, self-construal, and their interaction with state forgiveness were also examined using Model 1 in PROCESS macro ([27]). The results of the analysis predicting the state forgiveness with mindfulness (β = 0.27, t = −5.26, *p* < 0.001), self-construal index (β = 0.01, t = 0.30, *p* > 0.05), and their interaction (β = −0.10, t = 2.01, *p* = 0.045).

The interaction was further investigated using the procedures of [2] ([2]) by analyzing simple slopes for the two respects of self-construal and plotting them. As depicted in Figure 2, simple slopes were probed at 1 SD above and 1 SD below the mean on the self-construal index. Mindfulness was significantly positively related to state forgiveness at both low and high levels of the self-construal index; however, the slope was weaker at high levels (β = 0.16, t = 2.17, *p* < 0.001) compared to low levels (β = 0.37, t = 5.17, *p* = 0.031). These results indicate that interdependent self-construal contributes more to the positive correlation between mindfulness and forgiveness.

### 4.4. Prove the Hypothesis

#### 4.4.1. H1: Mindfulness Has a Positive Relationship with Trait and State Forgiveness

In this study, the H1 hypothesis was verified through descriptive statistics and correlation analysis; that is, mindfulness is positively correlated with trait forgiveness and state forgiveness. Specifically, the mean of mindfulness (MAAS) was 60.94, and the standard deviation was 10.27; the mean of trait forgiveness (TFS) was 25.37, and the standard deviation was 3.75; the mean of state forgiveness (TRIM-12) was 41.03, and the standard deviation was 7.35. Correlation analysis showed that the correlation coefficient between mindfulness and trait forgiveness was 0.18 (*p* < 0.01), and the correlation coefficient between mindfulness and state forgiveness was 0.26 (*p* < 0.01), both indicating that individuals with higher levels of mindfulness performed better in trait and state forgiveness. In addition, the correlation coefficient between trait forgiveness and state forgiveness was 0.35 (*p* < 0.01), further supporting the positive correlation between the two. The results of the common method bias analysis showed that the single factor variance explanation rate was 12.44%, which was lower than 40%, indicating that there was no serious common method bias in the data, ensuring the reliability of the correlation analysis results. These results provide strong empirical support for the positive correlation between mindfulness and forgiveness and verify the important role of mindfulness in promoting forgiveness.

#### 4.4.2. H2: Self-Construal Moderates the Relationship Between Mindfulness and Trait (State) Forgiveness

In this study, the data were analyzed using both correlation and moderation analysis, which supported the H2 hypothesis. That is to say, independent self-construal was found to be negatively correlated with trait forgiveness, while dependent self-construal was positively correlated with trait forgiveness. Specifically, the correlation coefficient between independent self-construal and trait forgiveness was found to be 0.12 (*p* < 0.05), indicating a significant positive correlation. However, the moderation analysis revealed that at a high level of independent self-construal, the relationship between mindfulness and trait forgiveness was not significant (β = 0. 09, t = 1.23, *p* > 0.05), while at a low level of independent self-construal, the relationship between mindfulness and trait forgiveness was significant (β = 0.30, t = 4.16, *p* < 0.001), which indicates that independent self-construal weakens the positive correlation between mindfulness and trait forgiveness to a certain extent. Conversely, the correlation coefficient between dependent self-construal and trait forgiveness was 0.20 (*p* < 0.01), signifying a significant positive correlation. The moderating effect analysis demonstrated that in cases of high levels of dependent self-construal, the relationship between mindfulness and trait forgiveness was significant, and the slope was substantial (β = 0.37, t = 5.17, *p* < 0.001). Conversely, at low levels of dependent self-construal, the relationship between mindfulness and trait forgiveness was also significant, but the slope was negligible (β = 0.16, t = 2.17, *p* < 0.001). This finding suggests that dependent self-construal amplifies the positive correlation between mindfulness and trait forgiveness. The findings of this study provide empirical evidence to support the hypothesis that self-construal plays a moderating role in the relationship between mindfulness and forgiveness. Furthermore, the results of this study verify the importance of dependent self-construal in promoting forgiveness.

## 5. Discussion

In the present study, the moderating effects of self-construal on the association between mindfulness and forgiveness are explored. The results obtained demonstrated that (a) trait mindfulness exhibited a positive association with both trait and state forgiveness, (b) independent and interdependent self-construal demonstrated a positive relationship with trait forgiveness, and (c) self-construal functioned as a moderator between mindfulness and trait and state forgiveness.

In accordance with the findings of previous studies ([68]; [37]), this study established positive correlations between mindfulness and both trait and state forgiveness. The mindfulness described in the hot/cool systems model of self-regulation is hypothesized to enhance forgiveness ([2]). Specifically, following an offense, individuals are hypothesized to be mentally activated in terms of their concrete, emotionally arousing “hot” features or their abstract, informational “cool” features. The focus on self-relevant events and emotions, typically in a state of self-immersed or egocentric, is a hallmark of this phenomenon. This “first-person” perspective can potentially result in the reliving of past experiences, leading to heightened negative arousal and compromised cognitive processing ability([38]). The practice of mindfulness enables individuals to identify and observe their experiences with greater clarity and objectivity, a process termed “reperceiving” ([62]). This process is analogous to decentering, automatization, and detachment, which are concepts that are fundamentally characterized by a shift in perspective ([62]). In contrast to the strong identification with and immersion in thoughts and emotions ([37]), individuals embodying “cool” features are better capable of processing an offensive incident in a more composed and reflective manner ([28]; [38]). This process entails the acknowledgment that much of the pain, depression, and thoughts are not intrinsic to the individual, thereby facilitating the abandonment of anger and the promotion of forgiveness. Consequently, mindfulness emerges as a pivotal factor in the forgiveness model.

The conclusion that both independent and interdependent self-construal are positively associated with trait forgiveness is consistent with previous findings ([71]). Forgiveness has been demonstrated to promote relational stability and harmony, which is consistent with the values of interdependent self-construal ([33]). Individuals who possess an interdependent self-construal are driven to forgive others and consequently act ([43]). However, it should be noted that the decisional forgiveness that is mediated by trait forgiveness may not be accompanied by emotional forgiveness ([43]; [32]). Consequently, individuals with interdependent self-construal forgive to enhance social harmony and repair relationships, while those with independent self-construal forgive to alleviate stress and enhance emotional well-being ([47]). This finding underscores the notion that both independent and interdependent self-construal are positively associated with trait forgiveness.

The present study investigated the role of self-construal as a moderator between mindfulness and forgiveness. Trait mindfulness exhibited a substantial positive predictive effect on both trait forgiveness and state forgiveness among participants characterized by increased interdependent self-construal; conversely, the predictive effects were weaker or non-significant among individuals with elevated independent self-construal. These findings are consistent with prior studies. [3] ([3]) found that mindfulness factors in self-kindness, common humanity, and self-compassion are positively correlated with interdependent self-construal. The studies indicate that mindfulness alters an individual’s self-perception, leading to a shift toward reduced egocentricity (a manifestation of independent self-construal) and an enhancement in prosocial behaviors ([22]). Interdependent self-construction, as a moderating factor, influences the relationship between mindfulness, cognitive reappraisal, and life satisfaction ([13]). The impact of mindfulness on prosocial behaviors is influenced by the degree of self-construal, with individuals exhibiting both independent and interdependent self-construal demonstrating varying responses to mindfulness ([55]). The practice of mindfulness has been shown to enhance self-awareness and clarify self-goals, irrespective of the degree of self-construal ([55]). Specifically, individuals with interdependent selves have a more intense tendency to forgive, whereas others with independent selves tend not to forgive in mindfulness. This finding, as empirical research, provides further support for the limited previous theory and research linking mindfulness with self-construal.

In collectivist cultures such as China, the primary mechanism for self-construction is dependent self-construction ([19]). This concept refers to the tendency of individuals to align their self-identity and behavior with those of others, placing significant emphasis on interpersonal relationships and the maintenance of social harmony. In this cultural context, mindfulness is associated not only with the maintenance of social relationships and group cohesion but also with the enhancement of interpersonal relationships and individual happiness, as well as social identity ([30]). The practice of mindfulness has been shown to enhance an individual’s propensity to forgive, particularly among those who exhibit dependent self-construal, a tendency to seek reconciliation and re-establish relationships. This forgiveness behavior has been demonstrated to contribute to the mitigation of social conflicts and the promotion of harmony within the group ([67]).

This study comprises a greater number of female subjects. This imbalance in gender distribution may result in the research findings reflecting a greater proportion of female characteristics and perspectives, potentially impacting the general applicability and external validity of the research conclusions. The samples originate from a single cultural background group and may not fully capture the diversity of cultural differences. Consequently, future research endeavors will focus on expanding the sample range to encompass a more diverse array of cultural backgrounds. This strategic decision is underpinned by the fundamental objective of enhancing the comprehensibility of psychological phenomena and behaviors across diverse cultural contexts. Through cross-cultural comparisons, the augmentation of the sample range will facilitate the acquisition of more nuanced perspectives and profound insights, thereby contributing to the advancement of global psychology research.

This study’s results also brought many insights. The theoretical revelation of this study is that it reveals the moderating role of self-construal in the relationship between mindfulness and forgiveness, adding new content to the theory of emotion regulation. The results demonstrate that the way an individual perceives themselves, that is, whether as an independent or dependent self-construal, significantly affects the predictive effect of mindfulness on forgiveness behavior. This provides a new perspective for understanding complex psychological processes and helps to build a more comprehensive psychological model. Furthermore, this study underscores the impact of cultural milieu on psychological constructs, particularly the predominance and significance of dependent self-construal within collectivist cultures. This provides an empirical foundation for the comparison of individual psychological distinctions across diverse cultural contexts, thereby contributing to the enhancement of cross-cultural psychology theory and the advancement of global psychology research in a more meticulous and comprehensive direction.

The practical revelation of this study is primarily evident in the domains of psychological intervention and education. In terms of psychological intervention, this study posits the notion that customized mindfulness training programs can be tailored to align with the distinctive characteristics of individual self-construal. For individuals with dependent self-construal, this study emphasizes the role of mindfulness in promoting harmonious interpersonal relationships. For those with an independent self-construal, the positive impact of mindfulness on self-awareness and goal-oriented behavior is highlighted, with the aim of more effectively enhancing the individual’s ability to forgive and improve their mental health. In the field of education, pedagogues can adopt different educational methods according to the self-construal tendencies of students. For example, group activities can be used to enhance the social belonging and forgiveness awareness of students with dependent self-construal. Alternatively, an autonomous learning space can be provided for independent self-construal students to improve their self-management ability, thereby creating a harmonious campus atmosphere. Furthermore, this study is instrumental in the promotion of cross-cultural communication and cooperation. By comprehending the variances in individual self-construal, individuals can enhance their capacity to predict and manage psychological conflicts in cross-cultural communication, mitigate cultural misunderstandings, and optimize the efficiency of communication and cooperation.

The present study is not without its limitations. Firstly, although this study examined the relationship between mindfulness, self-construal, and forgiveness, the use of a questionnaire as a measurement method is relatively simple and one-sided. For instance, previous research explored the association between forgiveness in different dimensions and self-construal ([46]; [69]). Utilizing the conceptual framework of interdependent and independent self-construal, researchers can manipulate offensive anchors (e.g., targeted at the individual, interpersonal relationships, or social norms). This approach serves to eliminate any potential correlation between forgiveness and self-construal. Secondly, the translation of psychological tendencies that differ from individual behavior into practical applications is challenging. Future research could consider conducting a behavioral experiment to verify the causality of these three variables. Thirdly, the theoretical and realistic model of mindfulness and self-construal has yet to be formulated. It is imperative to enhance this model to ensure its efficacy. A fourth consideration is the gender distribution of the research group, which is currently imbalanced. It is hypothesized that if the gender distribution were to be balanced, the research conclusions might change, with the correlation between mindfulness and forgiveness potentially weakening and the correlation between self-construction and forgiveness becoming apparent. In terms of moderating role, the promoting effect of mindfulness on forgiveness under men’s independent self-construction may weaken, and mindfulness practice under dependent self-construction may enhance men’s mediation ability. The attainment of gender balance is conducive to enhanced cross-cultural understanding of psychological constructs and provides a foundation for the development of psychological interventions tailored to different genders. Fifth, since the calculation of the moderator variable self-construction is obtained by subtracting the means of the two subscales, it is impossible to use the covariance method or partial least squares method to perform structural equation model analysis, which is also a limitation in the data analysis of this study.

Notwithstanding the limitations of this study, it provides valuable insights into the psychological mechanisms that underpin forgiveness and suggests practical applications for the promotion of prosocial behaviors and mental health in educational and clinical settings. By comprehending how self-construal moderates the relationship between mindfulness and forgiveness, psychologists and educators can formulate more efficacious strategies to cultivate forgiveness, enhance interpersonal relationships, and contribute to the overall well-being of individuals across diverse cultural contexts.

## 6. Conclusions

The present study provides evidence for the moderating role of self-construal in the relationship between mindfulness and forgiveness, highlighting the importance of considering cultural and individual differences in psychological interventions. The findings indicate that mindfulness is positively associated with both state and trait forgiveness. Independent self-construal is negatively related to trait forgiveness, whereas interdependent self-construal is positively related to trait forgiveness. This relationship is influenced by an individual’s self-construal, particularly in collectivist cultures where interdependent self-construal is more prevalent. This finding aligns with the hypothesis proposed. This study’s findings contribute to the understanding of how psychological interventions, such as mindfulness training, can be tailored to account for individual differences in self-construal to enhance forgiveness and, by extension, mental health and well-being. The observed moderation effect suggests that interventions may be more effective when they consider the cultural context and the individual’s self-construal style, emphasizing the need for a personalized approach in psychological education and therapy.

## Figures and Tables

**Figure 1 behavsci-15-00195-f001:**
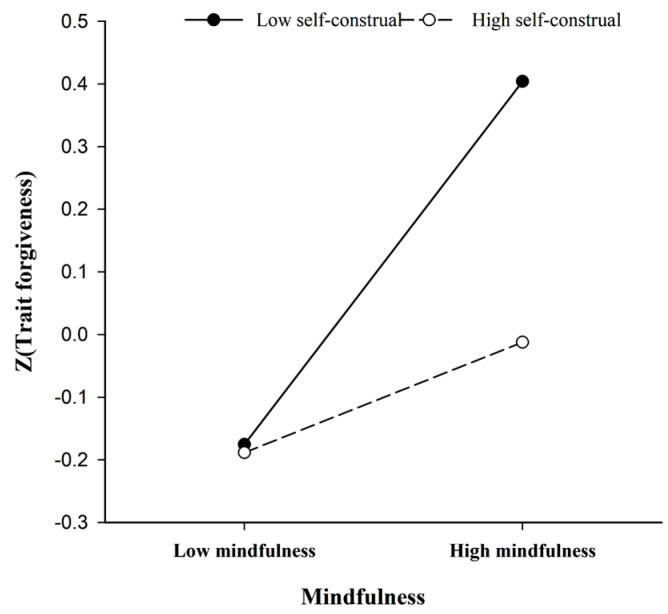
The moderating effect of self-construal on trait forgiveness and trait mindfulness.

**Figure 2 behavsci-15-00195-f002:**
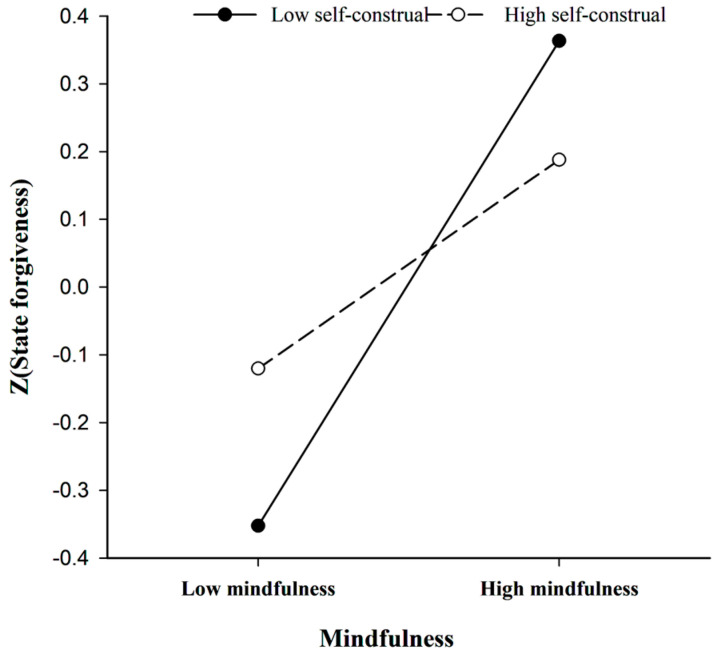
The moderating effect of self-construal on state forgiveness and trait mindfulness.

**Table 1 behavsci-15-00195-t001:** Bivariate associations and descriptive statistics (N = 408).

	M	SD	1	2	3	4	6	7
Trait mindfulness	60.94	10.27	1					
Trait forgiveness	25.37	3.75	0.18 **	1				
State forgiveness	41.03	7.35	0.26 **	0.35 **	1			
Self-construal	0.06	0.84	0.19 **	−0.06	0.08	1		
Independent self-construal	5.27	0.75	0.26 **	0.12 *	0.11 *	0.64 **	1	
Interdependent self-construal	25.21	0.68	0.05	0.20 **	0.03	−0.54 **	0.31 **	1

Note. * Correlation is significant at 0.05 (double-tailed). ** Correlation is significant at 0.01 (double-tailed).

## Data Availability

The data presented in this study are available on request from the corresponding author. The data are not publicly available due to their confidential contents, which could compromise the privacy of the research participants.

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
