# Peer review of "The Moderating Effect of Self-Construal on the Relationship Between Mindfulness and Forgiveness"

_behavsci, 2025, doi:10.3390/bs15020195_

Round 1

Reviewer 1 Report

Comments and Suggestions for Authors

Thank you for submitting your article. I have gone through your article with much interest. Even though, your research has some merit, there are major issues to be addressed.

The need for this research has not been clearly outlined as the research gaps have not been properly identified. You need to have a seperate section on literature review and the overarching theory that you are using for this study. There should be a seperate section for justifying the each hypothesis individually. In its current form, all the hypotheses are stated in the introduction section itself without proper theoretical argumentation. There is no clarity on how when and where the data was collected, who were your respondents, how the respondents were selected, etc. There is no description of the sample characteristics.

Importantly, the paper is seriously limited in data analysis. You should use latent variable structural equations modeling either covariance based or partial least squares based for analysing your data to validate the hypotheses. There is no information on the measurement properties of the constructs in your study. You need to report the reliability and validity of the constructs in your study.

Discussion needs to be improved in terms of the findings provide theoretical and practical implications. The conclusions section needs to improved, in its current form its just one sentence. 

The current form of your article has 34% similarity in ithenticate report which needs to be addressed. Some sentences look lifted directly from the source article. 

All the best to your research.

Comments on the Quality of English Language

The manuscript requires extensive proofreading. It appears that there are inappropriate phrases are used in a few places. For example, 108 line of the article mentions modulating role. It should have been moderating role. 

Author Response

To Reviewer:

Thank you for your letter and the constructive comments on this article in your busy schedule. All of our authors have  carefully read the comments that you have given us and have discussed and revised each of these issues.  Attached is a point-by-point reply to your suggestions. In addition, we have resubmitted a marked-up manuscript,  with the revisions highlighted in yellow. If there are any incorrect answers or questions in the manuscript,  please feel free to let us know.

Thank you again for your positive and constructive comments and suggestions on our manuscript. We hope you will find our revised manuscript acceptable for publication.

Best regards,

Yours sincerely,

Name: Haijiang Li

E-mail: haijiangli@shnu.edu.cn

Reviewer 2 Report

Comments and Suggestions for Authors

I have attached the file.

Author Response

(The authors gave the same response as above.)

Reviewer 3 Report

Comments and Suggestions for Authors

The study explores forgivness, and mindfulness empirically but the paper lacks a clear and unified explanation of how mindfulness, self-construal, and forgiveness are connected. Use some underlying theory to do so.  Is emiotion regulation somehow related to it. (2022). Emotion Regulation at Work Employees and Leaders’ Perspectives. International Journal of Innovation and Economic Development, 8(1), 50-71.

Some of the scales used have low reliability, which  weaken the accuracy of the results. The study relies entirely on self-reported questionnaires, which may introduce bias or inaccuracies. It only establishes correlations without proving any cause-and-effect relationships, as it does not include experiments or longitudinal analysis.

The sample is limited to young participants from a single cultural background, making it hard to generalize the findings to other populations. 

The model should be visually made, and include full questonnaire in the appendix.

Author Response

(The authors gave the same response as above.)

Round 2

Reviewer 1 Report

Comments and Suggestions for Authors

Dear Authors, Thank you for submitting the revised version of your manuscript. I am able to see slight improvement in comparison to previous version. However, many of my comments have not yet been addressed. The need for this research has not been clearly outlined as the research gaps have not yet  been properly identified. You need to have a separate section on literature review and the overarching theory that you are using for this study. There should be a separate section for justifying the each hypothesis individually. In this version, you removed the hypotheses statements, I do not know why. You need to have separate hypothesis justification for relationship between trait forgiveness and state forgiveness and also for moderation effects.  Your explanation on how when and where the data was collected, who were your respondents, how the respondents were selected, etc. are not sufficient. There is no description of the sample characteristics.

I am repeating my previous comments regarding data analysis: Importantly, the paper is seriously limited in data analysis. You should use latent variable structural equations modeling either covariance based or partial least squares based for analysing your data to validate the hypotheses. There is no information on the measurement properties of the constructs in your study. You need to report the reliability and validity of the constructs in your study.

I repeat my previous comments on discussion section: Discussion needs to be improved in terms of the findings provide theoretical and practical implications.

Reviewer 2 Report

Comments and Suggestions for Authors

I have attached it below. 

Round 3

Reviewer 1 Report

Comments and Suggestions for Authors

Thank you for addressing my comments. However, I am still not convinced why the authors are not able to perform partial least squares structural equations modeling using SmartPLS. This technique will be able to provide both convergent and discriminant validity results. This technique also allows for moderation testing appropriately. Without assessing the validity of the measures used in your study, I am not able to get convinced with the results of your study. 
